# Towards a Unified Theory of Learning and Information

**DOI:** 10.3390/e22040438

**Published:** 2020-04-13

**Authors:** Ibrahim Alabdulmohsin

**Affiliations:** Google Research, 8002 Zürich, Switzerland; ibomohsin@google.com

**Keywords:** statistical learning theory, information theory, entropy, parameter estimation, learning systems, privacy, prediction methods

## Abstract

In this paper, we introduce the notion of “learning capacity” for algorithms that learn from data, which is analogous to the Shannon channel capacity for communication systems. We show how “learning capacity” bridges the gap between statistical learning theory and information theory, and we will use it to derive generalization bounds for finite hypothesis spaces, differential privacy, and countable domains, among others. Moreover, we prove that under the Axiom of Choice, the existence of an empirical risk minimization (ERM) rule that has a vanishing learning capacity is equivalent to the assertion that the hypothesis space has a finite Vapnik–Chervonenkis (VC) dimension, thus establishing an equivalence relation between two of the most fundamental concepts in statistical learning theory and information theory. In addition, we show how the learning capacity of an algorithm provides important qualitative results, such as on the relation between generalization and algorithmic stability, information leakage, and data processing. Finally, we conclude by listing some open problems and suggesting future directions of research.

## 1. Introduction

### 1.1. Generalization Risk

A central goal when learning from data is to strike a balance between underfitting and overfitting. Mathematically, this requirement can be translated into an optimization problem with two competing objectives. First, we would like the learning algorithm to produce a hypothesis (i.e., an answer) that performs well on the empirical sample. This goal can be easily achieved by using a *rich* hypothesis space that can “explain” any observations. Second, we would like to guarantee that the performance of the hypothesis on the empirical data (a.k.a. training error) is a good approximation of its performance with respect to the unknown underlying distribution (a.k.a. test error). This goal can be achieved by *limiting* the complexity of the hypothesis space. The first condition mitigates underfitting while the latter condition mitigates overfitting.

Formally, suppose we have a learning algorithm $L:Z^{m}\to H$ that receives a sample $s=\{z_{1},\ldots ,z_{m}\}$, which comprises of *m* i.i.d. observations $z_{i}∼p\left(z\right)$, and uses s to select a hypothesis h∈H. Let *l* be a loss function defined on the product space Z×H. For instance, *l* can be the mean-square-error (MSE) in regression or the 0–1 error in classification. Then, the goal of learning from data is to select a hypothesis h∈H such that its *true risk*
R(h), defined by
(1)$$R\left(h\right)=E_{z∼p(z)}\left[l\left(z,h\right)\right],$$
is small. However, this optimization problem is often difficult to solve exactly since the underlying distribution of observations p(z) is seldom known. Rather, because the true risk R(h) can be decomposed into a sum of two terms:$$R\left(h\right)=\left[R_{s}\left(h\right)\right]+\left[R\left(h\right)-R_{s}\left(h\right)\right],$$
where $R_{s}\left(h\right)=E_{z∼s}\left[l\left(z,h\right)\right]≐\left(1/m\right)\sum_{z\in s}l\left(z,h\right)$, both terms can be tackled separately. The first term in the equation above corresponds to the *empirical risk* on the training sample s. The second term corresponds to the *generalization risk*. Hence, by minimizing both terms, one obtains a learning algorithm whose true risk is small.

Minimizing the empirical risk can be achieved using tractable approximations to the *empirical risk minimization* (ERM) procedure, such as stochastic convex optimization [1,2]. However, the generalization risk is often difficult to deal with directly because the underlying distribution is often unknown. Instead, it is a common practice to bound it *analytically*. By establishing analytical conditions for generalization, one hopes to design better learning algorithms that both perform well empirically and generalize as well into the future.

Several methods have been proposed in the past for bounding the generalization risk of learning algorithms. Some examples of popular approaches include uniform convergence, algorithmic stability, Rademacher and Gaussian complexities, and the PAC–Bayesian framework [3,4,5,6,7].

The proliferation of such bounds can be understood upon noting that the generalization risk of a learning algorithm is influenced by multiple factors, such as the domain Z, the hypothesis space H, and the mapping from Z to H. Hence, one may derive new generalization bounds by imposing conditions on any of such components. For example, the Vapnik–Chervonenkis (VC) theory derives generalization bounds by assuming constraints on H whereas stability bounds, e.g., [6,8,9], are derived by assuming constraints on the mapping from Z to H.

Rather than showing that certain conditions are sufficient for generalization, we will establish in this paper conditions that are both *necessary and sufficient*. More precisely, we will show that the “ uniform” generalization risk of a learning algorithm is an *information-theoretic* characterization. In particular, it is *equal* to the total variation distance between the joint distribution of the hypothesis h and a single random training example $\hat{z}∼s$, on one hand, and the product of their marginal distributions, on the other hand. Hence, it is analogous to the mutual information between h and $\hat{z}$. Since uniform generalization is an information-theoretic quantity, information-theoretic tools, such as the data-processing inequality and the chain rules of entropy [10], can be used to analyze the performance of machine learning algorithms. For example, we will illustrate this fact by presenting a simple proof to the classical generalization bound in the finite hypothesis space setting using, solely, information-theoretic inequalities without any reference to the union bound.

### 1.2. Types of Generalization

Generalization bounds can be stated either in expectation or in probability. Let l:Z×H→[0,1] be some loss function with a bounded range. Then, we have the following definitions:

**Definition** **1**(Generalization in Expectation)**.**
*The expected generalization risk of a learning algorithm*
$L:Z^{m}\to H$
*with respect to a loss*
l:Z×H→[0,1]
*is defined by:*
(2)$$R_{gen}\left(L\right)=E_{h}\left[R\left(h\right)\right]-E_{s,h}\;E_{\hat{z}∼s}\left[l\left(\hat{z},h\right)\right],$$
*where*
R(h)
*is defined in Equation (*Equation 1*), and the expectation is taken over the random choice of*
s
*and the internal randomness of*
L*. A learning algorithm*
L
*generalizes in expectation if*
$R_{gen}\left(L\right)\to 0$
*as*
m→∞
*for all distributions*
p(z)*.*

**Definition** **2**(Generalization in Probability)**.**
*A learning algorithm*
L
*generalizes in probability if for any*
ϵ>0*, we have:*
$$p\left\{|R\left(h\right)-E_{\hat{z}∼s}\left[l\left(\hat{z},h\right)\right]|>\epsilon \right\}\to 0\;\;\;\mathrm{as}\;m\to \infty ,$$
*where the probability is evaluated over the randomness of*
s
*and the internal randomness of the learning algorithm.*

In general, both types of generalization have been used to analyze machine learning algorithms. For instance, generalization in probability is used in the VC theory to analyze algorithms with finite VC dimensions, such as linear classifiers [3]. Generalization in expectation, on the other hand, was used to analyze learning algorithms, such as the stochastic gradient descent (SGD), differential privacy, and ridge regression [11,12,13,14]. Generalization in expectation is often simpler to analyze, but it provides a weaker performance guarantee.

### 1.3. Paper Outline

In this paper, a third notion of generalization is introduced, which is called *uniform* generalization. Uniform generalization also provides generalization bounds in expectation, but it is stronger than the traditional form of generalization in expectation in Definition 1 because it requires that the generalization risk vanishes uniformly in expectation across *all* bounded parametric loss functions (hence the name). In this paper, a loss function l:Z×H→[0,1] is called “ parametric” if it is conditionally independent of the original training sample given the learned hypothesis h∈H.

As mentioned earlier, the *uniform* generalization risk is *equal* to an information-theoretic quantity and it yields classical results in statistical learning theory. Perhaps more importantly, and unlike traditional in-expectation guarantees that do not imply concentration, we will show that uniform generalization in expectation implies generalization in probability. Hence, all of the uniform generalization bounds derived in this paper hold both in expectation and with a high probability.

The theory of uniform generalization bridges the gap between information theory and statistical learning theory. For example, we will establish an equivalence relation between the VC dimension, on one hand, and another quantity that is quite analogous to the Shannon channel capacity, on the other hand. Needless to mention, both the VC dimension and the Shannon channel capacity are arguably the most central concepts in statistical learning theory and information theory. This connection between the two concepts is obtained via the notion of the “ learning capacity” that we introduce in this paper, which is the supremum of the uniform generalization risk across all input distributions. We will compute the learning capacities for many machine learning algorithms and show how it matches known bounds on the generalization risk up to logarithmic factors.

In general, the main aim of this work is to bring to light a new information-theoretic approach for analyzing machine learning algorithms. Despite the fact that “ uniform generalization” might appear to be a strong condition at a first sight, one of the central themes that is emphasized repeatedly throughout this paper is that uniform generalization is, in fact, a natural condition that arises commonly in practice. It is not a condition to require or enforce by machine learning practitioners! We believe this holds because any learning algorithm is a *channel* from the space of training samples to the hypothesis space so its risk for overfitting can be analyzed by studying the properties of this mapping itself. Such an approach yields the uniform generalization bounds that are derived in this paper.

While we strive to introduce foundational results in this work, there are many important questions that remain unanswered. We conclude this paper by listing some of those open problems and suggesting future directions of research.

## 2. Notation

The notation used in this paper is fairly standard. Important exceptions are listed here. If x is a random variable that takes its values from a finite set s uniformly at random, we write x∼s to denote such a distribution. If x is a boolean random variable (i.e., a predicate), then I{x}=1 if and only if x is true, otherwise I{x}=0. In general, random variables are denoted with boldface letters x, instances of random variables are denoted with small letters *x*, matrices are denoted with capital letters *X*, and alphabets i.e., fixed sets) are denoted with calligraphic typeface X (except L that will be reserved for the learning algorithm and D that will be reserved for the input distribution as is customary in the literature).

Throughout this paper, we will always write Z to denote the space of observations (a.k.a. *domain*) and write H to denote the hypothesis space (a.k.a. *range*). A learning algorithm $L:\;Z^{m}\to H$ is formally treated as a stochastic map, where the hypothesis h∈H can be a deterministic or a randomized function of the training sample $s\in Z^{m}$. Given a 0–1 loss function l:H×Z→{0,1}, we will abuse terminology slightly by speaking about the “ VC dimension of H” when we actually mean the VC dimension of the loss class {l(·,h):h∈H}.

In addition, given two probability measures *p* and *q* defined on the same space, we will write 〈p,q〉 to denote the *overlapping coefficient* between *p* and *q*. That is, $\langle p,\;q\rangle =1-{\left||p\;,\;\;q|\right|}_{T}$, where ${\left||p\;,\;\;q|\right|}_{T}=\frac{1}{2}||p\;-\;q|{|}_{1}$ is the total variation distance.

Moreover, we will use the *order in probability* notation for real-valued *random* variables. Here, we adopt the notation used by [15] and [16]. In particular, let $x=x_{n}$ be a real-valued random variable that depends on some parameter n∈N. Then, we will write $x_{n}=O_{p}\left(f\left(n\right)\right)$ if for any δ>0, there exists absolute constants *C* and $n_{0}$ such that for any fixed $n\geq n_{0}$, the inequality $|x_{n}\left|<C\;|f\left(n\right)\right|$ holds with a probability of, at least, 1−δ. In other words, the ratio $x_{n}/f\left(n\right)$ is *stochastically* bounded [15]. Similarly, we write $x_{n}=o_{p}\left(f\left(n\right)\right)$ if $x_{n}/f\left(n\right)$ converges to zero in probability. As an example, if $x∼N(0,I_{d})$ is a standard multivariate Gaussian vector, then ${\left||\;x\;|\right|}_{2}=O_{p}\left(\sqrt{d}\right)$ even though ${\left||\;x\;|\right|}_{2}$ can be arbitrarily large. Intuitively, the probability of the event ${\left||\;x\;|\right|}_{2}\geq d^{\frac{1}{2}+\epsilon }$ when ϵ>0 goes to zero as d→∞ so ${\left||\;x\;|\right|}_{2}$ is *effectively* of the order $O(\sqrt{d})$.

## 3. Related Work

A learning algorithm is called *consistent* if the true risk of its hypothesis h converges to the optimal true risk in H, i.e., $\inf_{h\in H}R\left(h\right)$, as m→∞ in a distribution agnostic manner. A learning problem, which is a tuple (Z,H,l) with *l* being a loss function defined on the product space Z×H, is called *learnable* if it admits a consistent learning algorithm. It can be shown that learnability is equivalent to uniform convergence for supervised classification and regression even though uniform convergence is not necessary in the general setting [17].

Unlike learnability, the subject of generalization looks into how representative the empirical risk $R_{s}\left(h\right)$ is to the true risk R(h) as discussed earlier. It can be rightfully considered as an extension to the *law of large numbers*, which is one of the earliest and most important results in probability theory and statistics. However, unlike the law of large numbers, which assumes that observations are independent and identically distributed, the subject of generalization in machine learning addresses the case where the losses $l(z_{i},h)$ are no longer i.i.d. due to the fact that h is selected according to the training sample s and $z_{i}\in s$.

Similar to learnability, uniform convergence is, by definition, sufficient for generalization but it is not necessary because the learning algorithm might restrict its search space to a smaller subset of H. So, in addition to uniform convergence bounds, several other methods have been introduced for bounding the generalization risk, such as using algorithmic stability, Rademacher and Gaussian complexities, generic chaining bounds, the PAC-Bayesian framework, and robustness-based analysis [5,6,7,18,19,20]. Classical concentration of measure inequalities, such as using the union bound, form the building blocks of such rich theories.

In this work, we address the subject of generalization in machine learning from an information-theoretic point of view. We will show that if the hypothesis h conveys “ little” information about a random single training example $\hat{z}∼s$, then the difference between $E_{\hat{z}∼s}\left[l\left(\hat{z},h\right)\right]$ and $E_{z∼p(z)}\left[l\left(z,h\right)\right]$ will be small with a high probability. The measure of information we use here is given by the notion of *variational information*
$J(\hat{z};\;h)$ between the hypothesis h and a single random training example $\hat{z}∼s$. Variational information, also sometimes called *T*-information [14], is an instance of the class of *informativity* measures using *f*-divergences, which can be motivated axiomatically [21,22]. Unlike traditional methods, we will prove that $J(\hat{z};\;h)$ is *equal* to the “ uniform” generalization risk; it is not just an upper bound.

Information-theoretic approaches of analyzing the generalization risk of learning algorithms, such as the one proposed in this paper, have found applications in adaptive data analysis. This includes the work of [12] using the *max-information*, the work of [23] and [24] using the *mutual information*, and the work of [14] using the *leave-one-out* information. One key contribution of our work is to show that one should examine the relationship between the hypothesis and a *single* random training example, instead of examining the relationship between the hypothesis and the full training sample as is customary in the literature. The gap between such two approaches is strict. For example, Theorem 8 in Section 5.5 presents an example of when a learning algorithm can have a vanishing uniform generalization risk even when the mutual information between the learned hypothesis and the training sample can be made arbitrarily large.

## 4. Uniform Generalization

### 4.1. Preliminary Definitions

In this paper, we consider the general setting of learning introduced by Vapnik [3]. To reiterate, we have an observation space (a.k.a. domain) Z and a hypothesis space H. Our learning algorithm L receives a set of *m* observations $s=\left\{z_{1},\ldots ,z_{m}\right\}\in Z^{m}$ generated i.i.d. from some fixed unknown distribution p(z), and picks a hypothesis h∈H according to some probability distribution p(h|s). In other words, L is a channel from s to h. In this paper, we allow the hypothesis h to be any *summary statistic* of the training set. It can be an answer to a query, a measure of central tendency, or a mapping from the input space to the output space. In fact, we even allow h to be a subset of the training set itself. In formal terms, L is a stochastic map between the two random variables $s\in Z^{m}$ and h∈H, where the exact interpretation of those random variables is irrelevant. Moreover, we assume that there exists a non-negative bounded loss function l(z,h)∈[0,1] that is used to measure the fitness of the hypothesis h∈H on the observation z∈Z.

For any fixed hypothesis h∈H, we define its true risk R(h) by Equation (Equation 1) and denote its empirical risk on the training sample by $R_{s}\left(h\right)$. We also define the true and empirical risks of the *learning algorithm*
L by the expected corresponding risk of its hypothesis: (3)$$\begin{array}{cc} R(L) & =E_{s}\;E_{h\;∼p(h|s)}\;\left[R\left(h\right)\right]\;\;=E_{h}\;\left[R\left(h\right)\right] \end{array}$$(4)$$\begin{array}{cc} \hat{R}\left(L\right) & =E_{s}\;E_{h\;∼p(h|s)}\;\left[R_{s}\left(h\right)\right]\;=E_{s,\;h}\;\left[R_{s}\left(h\right)\right] \end{array}$$
Finally, the generalization risk of the learning algorithm is defined by:(5)$$R_{gen}\left(L\right)≐R\left(L\right)-\hat{R}\left(L\right)$$
Next, we define uniform generalization:

**Definition** **3**(Parametric Loss)**.**
*A loss function*
l(·,h):Z→[0,1]
*is called parametric if it is conditionally independent of the training sample given the hypothesis*
h∈H*. That is, it satisfies the Markov chain*
s→h→l(·,h)*.*

**Definition** **4**(Uniform Generalization)**.**
*A learning algorithm*
$L:\;Z^{m}\to H$
*generalizes uniformly with rate*
ϵ≥0
*if for all bounded parametric losses*
l:Z×H→[0,1]*, we have*
$|R_{gen}\left(L\right)|\leq \epsilon$*, where*
$R_{gen}\left(L\right)$
*is given in Equation (*Equation 5*).*

Informally, Definition 4 states that once a hypothesis h is selected by a learning algorithm L that achieves uniform generalization, then no “ adversary” can post-process the hypothesis in a manner that causes over-fitting to occur. Equivalently, uniform generalization implies that the empirical performance of h on the sample s will remain close to its performance with respect to the underlying distribution regardless of how that performance is being measured. For example, the loss function l:Z×H→[0,1] in Equation (Equation 5) can be the misclassification error rate as in the traditional classification setting, a cost-sensitive error rate as in fraud detection and medical diagnosis [25], or the Brier score as in probabilistic predictions [26]. The generalization guarantee would hold in any case.

### 4.2. Variational Information

Given two random variables x and y, the *variational information* between the two random variables is defined to be the total variation distance between the join distribution p(x,y) and the product of marginals p(x)·p(y). We will denote this by J(x;y). By definition:$$J\left(x;\;y\right)=E_{x,y}||p\left(x,y\right)\;,\;\;p\left(x\right)\cdot {p\left(y\right)||}_{T}=E_{x}{\left||p\left(y\right)\;,\;\;p\left(y|x\right)|\right|}_{T}$$
Note that 0≤J(x;y)≤1. We describe some of the important properties of variational information in this section. The reader may consult the appendices for detailed proofs.

**Lemma** **1**(Data Processing Inequality)**.**
*If*
x→y→z
*is a Markov chain, then:*
J(x;z)≤J(y;z)

This *data processing inequality* holds, in general, for all informativity measures using *f*-divergences [21,22].

**Lemma** **2**(Information Cannot Hurt)**.**
*For any random variables*
x∈X*,*
y∈Y*, and*
z∈Z*, we have:*
J(x;y)≤J(x;(y,z))

**Proof.** The proof is in Appendix A. □

Finally, we derive a chain rule for the variational information.

**Definition** **5**(Conditional Variational Information)**.**
*The conditional variational information between the two random variables*
x
*and*
y
*given*
z
*is defined by:*
$$J\left(x;\;y\;|\;z\right)=E_{z}\left[||p\left(x,y\;|\;z\right)\;,\;\;p\left(x|z\right)\cdot {p\left(y|z\right)||}_{T}\right],$$
*which is analogous to the conditional mutual information in information theory [*[10]*].*

**Theorem** **1**(Chain Rule)**.**
*Let*
$\left(h_{1},\ldots ,h_{k}\right)$
*be a sequence of random variables. Then, for any random variable*
z*, we have:*
$J\left(z;\;\left(h_{1},\ldots ,h_{k}\right)\right)\leq \sum_{t=1}^{k}J\left(z;\;h_{t}\;|\;\left(h_{1},\ldots ,h_{t-1}\right)\right)$

**Proof.** The proof is in Appendix B. □

Although the chain rule above provides an upper bound, the upper bound is tight in the following sense:

**Proposition** **1.**
*For any random variables*
x,y
*, and*
z
*, we have*
|J(x;(y,z))−J(x;z|y)|≤J(x;y)
*and*
|J(x;(y,z))−J(x;y)|≤J(x;z|y)
*.*


**Proof.** The proof is in Appendix C. □

In other words, the inequality in the chain rule J(x;(y,z))≤J(x;y)+J(x;z|y) becomes an equality if:min{J(x;y),J(x;z|y)}=0

The chain rule provides a recipe for computing the bias of a composition of hypotheses $\left(h_{1},\ldots ,h_{k}\right)$. Recently, [23] proposed an *information budget* framework for controlling the bias of estimators by controlling the mutual information between h and the training sample s. The proposed framework rests on the chain rule of mutual information. Here, we note that the argument for the information budget framework also holds when using the variational information due to the chain rule above.

### 4.3. Equivalence Result

Our first main theorem states that the uniform generalization risk has a precise information-theoretic characterization.

**Theorem** **2.**
*Given a fixed constant*
0≤ϵ≤1
*and a learning algorithm*
$L:\;Z^{m}\to H$
*that selects a hypothesis*
h∈H
*according to a training sample*
$s=\{z_{1},\ldots ,z_{m}\}$
*, where*
$z_{i}∼p\left(z\right)$
*are i.i.d.,*
L
*generalizes uniformly with rate ϵ if and only if*
$J(h;\;\hat{z})\leq \epsilon$
*, where*
$\hat{z}∼s$
*is a single random training example.*


**Proof.** Let $L:Z^{m}\to H$ be a learning algorithm that receives a finite set of training examples $s=\left\{z_{1},\ldots ,z_{m}\right\}\in Z^{m}$ drawn i.i.d. from a fixed unknown distribution p(z). Let h∼p(h|s) be the hypothesis chosen by L (can be deterministic or randomized) and write $\hat{z}∼s$ to denote a random variable that selects its value uniformly at random from the training sample s. Clearly, $\hat{z}$ and h are not independent in general. To simplify notation, we will write l=l(·,h):Z→[0,1] to denote the loss function. Note that l is itself a random variable that satisfies the Markov chain s→h→l. The claim is that L generalizes uniformly with rate ϵ>0 across all parametric loss functions l if and only if $J(h;\;\hat{z})\leq \epsilon$.By the Markov property, we have p(l|h,s)=p(l|h). By definition, the true and empirical risks of L are given by:
(6)$$\begin{array}{cc} R(L) & =E_{s,h}\;E_{l|h}E_{z∼p(z)}\;l\left(z\right)=E_{l}\;E_{z∼p(z)}\;l\left(z\right) \end{array}$$
(7)$$\begin{array}{cc} \hat{R}\left(L\right) & =E_{s}\;E_{l|s}\;E_{z∼s}\;l\left(z\right)=E_{l}\;E_{s|l}\;E_{z∼s}\;l\left(z\right) \end{array}$$
Because $\hat{z}∼s$ is a random variable whose value is chosen uniformly at random with replacement from the training set s, its marginal distribution is p(z). Its *conditional* distribution given l can be different, however, because both l and $\hat{z}$ depend on the training set s. However, they are both *conditionally* independent of each other given s. By marginalization, we have:
$$p\left(\hat{z}|l\right)=E_{s|l}\;p\left(\hat{z}|s,l\right)=E_{s|l}\;p\left(\hat{z}|s\right)$$
Combining this with Equations (Equation 6) and (7) yields $R\left(L\right)=E_{l}\;E_{\hat{z}}\;l\left(\hat{z}\right)$ and $\hat{R}\left(L\right)=E_{l}\;E_{\hat{z}|l}\;l\left(\hat{z}\right)$. Both equations imply that:
$$R\left(L\right)-\hat{R}\left(L\right)=E_{l}\;\left[E_{\hat{z}}\;l\left(\hat{z}\right)-E_{\hat{z}|l}\;l\left(\hat{z}\right)\right]$$
Now, we would like to sandwich the right-hand side between upper and lower bounds. To do this, we note that if $p_{1}\left(z\right)$ and $p_{2}\left(z\right)$ are two distributions defined on the same domain Z and f:Z→[0,1], then:
$$|E_{z∼p_{1}\left(z\right)}\;f\left(z\right)-E_{z∼p_{2}\left(z\right)}\;f\left(z\right)|\leq \left|\right|p_{1}\left(z\right)\;,\;\;p_{2}\left(z\right){\left|\right|}_{T},$$
where $\left|\right|p_{1}\left(z\right)\;,\;\;p_{2}\left(z\right){\left|\right|}_{T}$ is the total variation distance. This result can be immediately proven by considering the two regions $\left\{z\in Z:\;p_{1}\left(z\right)>p_{2}\left(z\right)\right\}$ and $\left\{z\in Z:\;p_{1}\left(z\right)<p_{2}\left(z\right)\right\}$ separately. In addition, it is tight because the inequality holds with equality for the loss function $f\left(z\right)=I\{p_{1}\left(z\right)\geq p_{2}\left(z\right)\}$. Consequently:
$$|R\left(L\right)-\hat{R}\left(L\right)|\leq J\left(l;\;\hat{z}\right)$$
Finally, from the Markov chain $\hat{z}\to s\to h\to l$ and the data processing inequality, we have $J\left(l;\;\hat{z}\right)\leq J\left(h;\;\hat{z}\right)$. Plugging this into the earlier inequality yields the bound:
$$|R\left(L\right)-\hat{R}\left(L\right)|\leq J\left(h;\;\hat{z}\right)$$To prove the converse, define:
$$\begin{array}{cc} l^{⋆}\left(z,h\right) & =I\left\{p\left(\hat{z}=z\right)\;\geq p\left(\hat{z}=z\;|\;h\right)\right\}\\ \; & =I\left\{p\left(\hat{z}=z\right)\;\geq \;E_{s\;|\;h}\;\left[p_{\hat{z}∼s}\;\left(\hat{z}=z\right)\right]\right\} \end{array}$$The loss $l^{⋆}\left(z,h\right)$ is independent of the training sample given h because $p(\hat{z}=z\;|\;h)$ is evaluated by taking expectation over all the training samples conditioned on h. Hence, $l^{⋆}\left(z,h\right)$ is a 0–1 loss defined on the product space Z×H and satisfies the Markov chain s→h→l. However, given this choice of loss, we have:
$$\begin{array}{cc} |R\left(L\right)-\hat{R}\left(L\right)| & =E_{h}\left[E_{\hat{z}}\;I\left\{p\left(\hat{z}\right)>p\left(\hat{z}\;|\;h\right)\right\}-E_{\hat{z}\;|\;h}\;I\left\{p\left(\hat{z}\right)>p\left(\hat{z}\;|\;h\right)\right\}\right]\\ \; & =E_{h}||p\left(\hat{z}\right)\;,\;\;p\left(\hat{z}\;|\;h\right){\left|\right|}_{T}=J\left(h;\;\hat{z}\right) \end{array}$$Hence, the variational information $J(h;\;\hat{z})$ does not only provide an upper bound on the uniform generalization risk, but is also a lower bound to it. Therefore, $J(h;\;\hat{z})$ is equal to the uniform generalization risk. □

**Remark** **1.***One important observation about Theorem 2 is that the variational information is measured between the hypothesis*h*and a* single *training example*
$\hat{z}$*, which is quite different from previous works that looked into the mutual information with the entire training sample*
s*. By considering*
$\hat{z}$
*rather than*
s*, we quantify the uniform generalization risk with equality and the resulting bound is not vacuous even if the learning algorithm was deterministic. By contrast,*
J(s;h)
*may yield vacuous bounds when*
L
*is deterministic and both*
Z
*and*
H
*are uncountable.*

For concreteness, we illustrate how to compute the uniform generalization risk (or equivalently the variational information) on two simple examples. Here, B(k;ϕ,n)=nkϕk(1−ϕ)n−k is the binomial distribution. The first example is a special case of a more general theorem that will be presented later in Section 5.2.

**Example** **1.**
*Suppose that observations*
$z_{i}\in \left\{0,\;1\right\}$
*are i.i.d. Bernoulli trials with*
$p(z_{i}=1)=\phi$
*, and that the hypothesis produced by*
L
*is the empirical average*
$h=\frac{1}{m}\;\sum_{i=1}^{m}\;z_{i}$
*. Because*
$p\left(h=k/m\;|\;z_{trn}=1\right)=B\left(k-1;\phi ,m-1\right)$
*and*
$p\left(h=k/m\;|\;z_{trn}=0\right)=B\left(k;\phi ,m-1\right)$
*, it can be shown that the uniform generalization risk of this learning algorithm is given by the following quantity assuming that*
ϕm
*is an integer:*
(8)J(z^;h)=2(1−ϕ)(1−ϕ)mϕ1+mϕ(1+mϕ)mmϕ+1
*This is maximized when*
ϕ=1/2
*, in which case, the uniform generalization risk can be bounded using the Stirling approximation [*
[27]
*] by*
$1/\sqrt{2\;\pi \;m}$
*up to a first-order term.*


**Proof.** First, the probability we obtain a hypothesis $h=\frac{k}{m}$, where k∈{0,1,…,m}, given that we have *m* Bernoulli trials has a binomial distribution:
p(h=km)=mkϕk(1−ϕ)m−k
We use the identity:
$$J\left(\hat{z};\;h\right)=\sum_{k=0}^{m}\;p\left(h=\frac{k}{m}\right)\;||p\left(\hat{z}\right)\;,\;\;p\left(\hat{z}|h\right){\left|\right|}_{T}$$However, $p(\hat{z})$ is Bernoulli with probability of success ϕ while $p(\hat{z}|h=\frac{k}{m})$ is Bernoulli with probability of success h. The total variation distance between the two Bernoulli distributions is given by |ϕ−h|. So, we obtain:
(9)J(z^;h)=∑k=0mmkϕk(1−ϕ)m−k|ϕ−km|
This is the *mean deviation*. Assuming ϕm is an integer, then the mean deviation of the binomial random variable is given by de Moivre’s formula:
(10)MD=2(1−ϕ)(1−ϕ)mϕ1+mϕ(1+mϕ)mmϕ+1
The mean deviation is maximized when $\phi =\frac{1}{2}$. This gives us:
J(z^;h)≤12mmm/2+1∼12πm,
where in the last step we expanded the binomial coefficient and used Stirling’s approximation [27]. □

**Example** **2.**
*Suppose that the domain is*
Z={1,2,3,…,K}
*for some*
K<∞
*, where*
p(z=k)=1/K
*for all*
k∈Z
*. Let the hypothesis space be*
H=Z
*where*
p(h=k)
*is equal to the fraction of times the value k is observed in the training sample*
$s=\{z_{1},\ldots ,z_{m}\}$
*. For example, if*
s={1,3,2,1,1,3}
*, the hypothesis*
h
*is chosen among the set*
{1,2,3}
*with the respective probabilities*
{1/2,1/6,1/3}
*. Then, the variational information is given by:*
$$J\left(\hat{z};\;h\right)=\frac{1}{m}\left(1-\frac{1}{K}\right)$$


**Proof.** We have by symmetry p(h=k)=1/K for all k∈{1,2,3,…,K}. Let $\hat{z}=x$. By Bayes rule, we have:
$$\begin{array}{cc} p(\hat{z}=x|h=k) & =p\left(h=k|\hat{z}=x\right)\cdot \frac{p(\hat{z}=x)}{p(h)=k}\\ \; & =p(h=k|\hat{z}=x) \end{array}$$
However, given one observation $\hat{z}=x$, the probability of selecting a hypothesis h=k depends on two cases:
$$p\left(h=k\;|\;\hat{z}=x\right)=\left\{\begin{array}{cc} q & \mathrm{if}\;k=x\\ r & \mathrm{if}\;k\neq x \end{array}\right.$$
for some values q≥0 and r≥0 such that q+(K−1)r=1. To find *q*, we use the definition of L:
$$q=\frac{1}{m}+\frac{1}{K}\cdot \frac{m-1}{m}=\frac{1}{K}+\frac{1}{m}\left(1-\frac{1}{K}\right)$$
This holds because L is equivalent to an algorithm that selects a single observation in the set s uniformly at random. So, to satisfy the condition q+(K−1)r=1, we have:
$$r=\frac{1}{K}-\frac{1}{mK}$$
Now, we are ready to find the desired expression.
$$\begin{array}{cc} J(\hat{z};\;h) & =\frac{1}{2}\sum_{x\in Z}p\left(\hat{z}=x\right)\sum_{k\in Z}|p\left(h=k\right)-p\left(h=k|\hat{z}=x\right)|\\ \; & =\frac{1}{2}\sum_{k\in Z}|p\left(h=k\right)-p\left(h=k|\hat{z}=1\right)|\\ \; & =\frac{1}{2}\left[\frac{1}{m}\left(1-\frac{1}{K}\right)+\frac{K-1}{mK}\right]=\frac{1}{m}\left(1-\frac{1}{K}\right) \end{array}$$ □

Note that the variational information in Example 2 is Θ(1/m), which is smaller than the variational information in Example 1. This is not a coincidence. The difference between the two examples is related to *data processing*. Specifically, suppose that K=2 in Example 2 and let $h_{2}$ be the hypothesis. Let $h_{1}$ be the hypothesis in Example 1. Then, we have the Markov chain $s\to h_{1}\to h_{2}$ because $h_{2}$ is Bernoulli with parameter $h_{1}$.

### 4.4. Learning Capacity

The variational information depends on the distribution of observations p(z), which is seldom known in practice. To construct a distribution-free bound on the uniform generalization risk, we introduce the following quantity:

**Definition** **6**(Learning Capacity)**.**
*The learning capacity of an algorithm*
L
*is defined by:*
(11)$$C\left(L\right)≐\underset{p(z)}{\sup}\;\left\{J(\hat{z};\;h)\right\},$$
*where*
h
*and*
$\hat{z}$
*are as defined in Theorem 2.*

The above quantity is analogous to the Shannon channel capacity except that it is measured in the total variation distance. It quantifies the capacity for overfitting in the given learning algorithm. For example, the learning capacity of the algorithm in Example 1 is $1/\sqrt{2\pi m}$ up to a first order term, as proved earlier, so its capacity for overfitting is larger than that of the learning algorithm in Example 2.

Theorem 2 reveals that C(L) has, at least, three *equivalent* interpretations:*Statistical*: The learning capacity C(L) is equal to the supremum of the expected generalization risk $R_{gen}\left(L\right)$ across all input distributions and all bounded parametric losses. This holds by Theorem 2 and Definition 6.*Information-Theoretic*: The learning capacity C(L) is equal to the amount of information contained in the hypothesis h about the training examples. This holds because $J\left(\hat{z};\;h\right)=E_{h}||p\left(\hat{z}\right)\;,\;\;p\left(\hat{z}\;|\;h\right){\left|\right|}_{T}$.*Algorithmic*: The learning capacity C(L) measures the influence of a single training example $\hat{z}$ on the distribution of the final hypothesis h. As such, a learning algorithm has a small learning capacity if and only if it is algorithmically stable. This follows from the fact that $J\left(\hat{z};\;h\right)=E_{\hat{z}}||p\left(h\right)\;,\;\;p\left(h\;|\;\hat{z}\right){\left|\right|}_{T}$.

Throughout the sequel, we analyze the properties of C(L) and derive upper bounds for it under various conditions, such as in the finite hypothesis space setting and differential privacy.

### 4.5. The Definition of Hypothesis

In the proof of Theorem 2, the following Markov chain $\hat{z}\to s\to h\to l\left(\cdot ,h\right)$ is used. Essentially, this states that the loss function l(·,h):Z→[0,1], which is a random variable itself, must be parameterized entirely by the hypothesis h as stated in Definition 3. We list, next, a few examples that highlight this point.

**Example** **3**(Input Normalization)**.**
*If the data is normalized prior to training, such as using min-max or z-score normalization, then the normalization parameters are included in the definition of the hypothesis*
h*.*

**Example** **4**(Feature Selection)**.**
*If the observations*
z
*comprise of d features and feature selection is implemented prior to training a model*
v
*(such as in classification or clustering), then the hypothesis*
h
*is the composition*
(u,v)*, where*
$u\in {\left\{0,1\right\}}^{d}$
*encodes the set of the features that have been selected by the feature selection algorithm.*

**Example** **5**(Cross Validation)**.**
*Hyper-parameter tuning is a common practice in machine learning. This includes choosing the tradeoff parameter C in support vector machine (SVM) [*[28]*] or the bandwidth γ in radial basis function (RBF) networks [*[29]*]. However, not all hyper-parameters are encoded in the hypothesis*
h*. For instance, the tradeoff constant C is never used during prediction so it is omitted from the definition of*
h
*but the bandwidth parameter γ is included if it is selected based on the training sample.*

In order to illustrate why the Markov chain $\hat{z}\to s\to h\to l\left(\cdot ,h\right)$ is important, consider the following simple scenario. Suppose we have a mixture of two Gaussians in $R^{d}$, one corresponding to the positive class and one corresponding to the negative class. If *z*-score normalization is applied before training a linear classifier, then the generalization risk might increase with normalization because the final hypothesis now includes more information about the training sample (see Lemma 2). Figure 1 shows this effect when d=1. As illustrated in the figure, normalization is often important in order to assign equal weights to all features but it can increase the generalization risk as well.

### 4.6. Concentration

The notion of uniform generalization in Definition 4 provides *in-expectation* guarantees. In this section, we show that whereas traditional generalization in expectation does not imply concentration, *uniform* generalization in expectation implies concentration. In fact, we will use the chain rule in Theorem 1 to derive a Markov-type inequality. After that, we show that the bound is tight.

We begin by showing why a non-uniform generalization in expectation does not imply concentration.

**Proposition** **2.**
*There exists a learning algorithm*
$L:\;Z^{m}\to H$
*and a parametric loss*
l:Z×H→[0,1]
*such that the expected generalization risk is*
$R_{gen}\left(L\right)=0$
*even though*
$p\left\{|R\left(h\right)-R_{s}\left(h\right)|=\frac{1}{2}\right\}=1$
*, where the probability is evaluated over the randomness of*
s
*and the internal randomness of*
L
*.*


**Proof.** Let Z=[0,1] be an instance space with a continuous marginal density p(z) and let Y={−1,+1} be the target set. Let $h^{⋆}:Z\to \left\{-1,+1\right\}$ be some *fixed* predictor, such that $p\left\{h^{⋆}\left(z\right)=1\right\}=\frac{1}{2}$, where the probability is evaluated over the random choice of z∈Z. In other words, the marginal distribution of the labels predicted by $h^{⋆}$ is uniform over the set {−1,+1}. These assumptions are satisfied, for example, if p(z) is uniform in [0,1] and $h^{⋆}\left(z\right)=I\left\{z<1/2\right\}$.Next, let the hypothesis space H be the set of predictors from Z to {−1,+1} that output a label in {−1,+1} uniformly at random everywhere in Z except at a finite number of points. Define the parametric loss by $l\left(z;h\right)=I\left\{h\left(z\right)\neq h^{⋆}\left(z\right)\right\}$.Next, we construct a learning algorithm L that generalizes perfectly in expectation but does not generalize in probability. The learning algorithm L simply picks $h\in \{h_{0},h_{1}\}$ at random with equal probability. The two hypotheses are:
$$\begin{array}{cc} h_{0}\left(z\right) & =\left\{\begin{array}{cc} -h^{⋆}\left(z\right) & \;\mathrm{if}\;z\in s\\ \mathrm{Uniform}(-1,+1) & \;\mathrm{if}\;z\notin s \end{array}\right.\\ h_{1}\left(z\right) & =\left\{\begin{array}{cc} h^{⋆}\left(z\right) & \;\mathrm{if}\;z\in s\\ \mathrm{Uniform}(-1,+1) & \;\mathrm{if}\;z\notin s \end{array}\right. \end{array}$$
Because Z is uncountable, where the probability of seeing the same observation z twice is zero, $R\left(h\right)=\frac{1}{2}$ for this learning algorithm. Thus:
$$R_{gen}\left(L\right)=E_{s,h}\left[R_{s}\left(h\right)-R\left(h\right)\right]=0$$
However, the empirical risk for any s satisfies $R_{s}\left(h\right)\in \left\{0,1\right\}$ while the true risk always satisfies $R\left(h\right)=\frac{1}{2}$, as mentioned earlier. Hence, the statement of the proposition follows. □

There are many ways of seeing why the algorithm in Proposition 2 does not generalize *uniformly* in expectation. The simplest way is to use the equivalence between uniform generalization and variational information as stated in Theorem 2. Given the hypothesis $h\in \{h_{0},h_{1}\}$ that is learned by the algorithm constructed in the proposition, the marginal distribution of an individual training example $p(\hat{z}\;|\;h)$ is uniform over the sample s. This follows from the fact that the hypothesis h has to encode the entire sample s. However, the probability of seeing the same observation twice is zero (by construction). Hence, $||p\left(\hat{z}\right)\;,\;\;p\left(\hat{z}\;|\;h\right){\left|\right|}_{T}=1$. This shows that C(L)=1.

The example in Proposition 2 reveals an interesting property of non-uniform generalization. Namely, *non-uniform* generalization can be sensitive to every bit of information provided by the hypothesis. In the example above, the hypothesis h is encoded by the pair (s,k), where k∈{0,1} determines which of the two hypotheses $\left\{h_{0},h_{1}\right\}$ is selected. The discrepancy between generalization in expectation and generalization in probability happens because k is added into the hypothesis.

Next, we use the chain rule in Theorem 1 to prove that uniform generalization, on the other hand, is a *robust* property of learning algorithms. More precisely, if k has a finite domain, then a hypothesis h generalizes uniformly in expectation if and only if the pair (h,k) generalizes uniformly in expectation. Hence, adding any finite amount of information (in bits) to a hypothesis cannot alter its uniform generalization property in a significant way.

**Theorem** **3.**
*Let*
$L:\;Z^{m}\to H$
*be a learning algorithm whose hypothesis is*
h∈H
*. Let*
k∈K
*be a different hypothesis that is obtained from the same sample*
s
*. If*
$\hat{z}∼s$
*, then:*
$$\begin{array}{c} J\left(\hat{z};\;\left(h,k\right)\right)\leq \left(2+\frac{\left|K\right|}{2}\right)\cdot J\left(\hat{z};\;h\right)+\sqrt{\frac{\log|K|}{2m}} \end{array}$$


**Proof.** The proof is in Appendix D. □

We use Theorem 3, next, to prove that a uniform generalization in expectation implies a generalization in probability. The proof is by contradiction. Suppose we have a hypothesis h that generalizes uniformly in expectation but there exists a parametric loss l:Z×H→[0,1] that does not generalize in probability. We will derive a contradiction from these two assumptions. We show that appending little information to the hypothesis h will allow us to construct a *different* parametric loss that does not generalize in expectation by determining whether or not the empirical risk w.r.t. l:Z×H→[0,1] is greater than, approximately equal to, or is less than the true risk w.r.t. the same loss. This is described in, at most, two bits. Knowing this additional information, we can define a new parametric loss that does not generalize in expectation, which contradicts the definition of uniform generalization.

**Theorem** **4.**
*Let*
$L:\;Z^{m}\to H$
*be a learning algorithm, whose risk is evaluated using a parametric loss*
l:Z×H→[0,1]
*. Then:*
$$\begin{array}{c} p\left\{|R_{s}\left(h\right)-R\left(h\right)|\geq t\right\}\leq \frac{7}{2t}\left[J\left(\hat{z};\;h\right)+\sqrt{\frac{\log3}{49m}}\right], \end{array}$$
*where the probability is evaluated over the random choice of*
s
*and the internal randomness of*
L
*.*


**Proof.** Let l:Z×H→[0,1] be a parametric loss function and write:
(12)$$\kappa \left(t\right)=p\left\{|R_{s}\left(h\right)-R\left(h\right)|\geq t\right\}$$
Consider the new pair of hypotheses (h,k), where:
$$k=\left\{\begin{array}{cc} +1, & \;\mathrm{if}\;R_{s}\left(h\right)\geq R\left(h\right)+t\\ -1, & \;\mathrm{if}\;R_{s}\left(h\right)\leq R\left(h\right)-t\\ 0, & \mathrm{otherwise} \end{array}\right.$$
Then, by Theorem 3, the uniform generalization risk in expectation for the composition of hypotheses (h,k) is bounded by $\left(7/2\right)\;J\left(\hat{z};\;h\right)+\sqrt{\frac{\log3}{2m}}$. This holds uniformly across all parametric loss functions that satisfy the Markov chain s→(h,k)→l(·,(h,k)). Next, consider the parametric loss:
$$l\left(z,\left(h,k\right)\right)=\left\{\begin{array}{cc} l(z;h) & \;\mathrm{if}\;k=+1\\ 1-l(z;h) & \;\mathrm{if}\;k=-1\\ 0 & \;\mathrm{otherwise} \end{array}\right.$$
Note that l(z,(h,k)) is parametric with respect to the composition of hypotheses (h,k). Using Equation (Equation 12), the generalization risk w.r.t l(z,(h,k)) in expectation is, at least, as large as tκ(t). Therefore, by Theorems 2 and 3, we have $t\;\kappa \left(t\right)\leq \left(7/2\right)\;J\left(\hat{z};\;h\right)+\sqrt{\frac{\log3}{2m}}$, which is the statement of the theorem (Note: The proof assumes that the loss function l has access to the underlying distribution. This assumption is valid because the underlying distribution p(z) is fixed and does not depend on any random outcomes, such as s or h). □

Theorem 4 reveals that uniform generalization is sufficient for concentration to hold. Importantly, the generalization bound depends on the learning algorithm L only via its variational information $J(\hat{z};\;h)$. Hence, by controlling the uniform generalization risk, one improves the generalization risk of L both in expectation and with a high probability.

The same proof technique used in Theorem 4 also implies the following concentration bound, which is useful when I(h;s)=o(m) where I(x;y) is the Shannon mutual information. The following bound is similar to the bound derived by [23] using properties of sub-Gaussian loss functions.

**Proposition** **3.**
*Let*
$L:\;Z^{m}\to H$
*be a learning algorithm, whose risk is evaluated using a parametric loss function*
l:Z×H→[0,1]
*. Then:*
$$\begin{array}{c} p\left\{|R_{s}\left(h\right)-R\left(h\right)|\geq t\right\}\leq \frac{1}{t}\sqrt{\frac{I(s;h)+2}{2m}}. \end{array}$$


**Proof.** The proof is in Appendix E. □

Note that having a vanishing mutual information, i.e., I(s;h)=o(m), which is the setting recently considered in the work of [23], is a *strictly stronger* condition than uniform generalization. For instance, we will later construct *deterministic* learning algorithms that generalize uniformly in expectation even though I(s;h) is unbounded (see Theorem 8). By contrast, I(s;h)=o(m) is sufficient for $J(\hat{z};\;h)\to 0$ to hold.

Finally, we note that the concentration bound depends linearly on the variational information $J(\hat{z};\;h)$. Typically, $J\left(\hat{z};\;h\right)=O\left(1/\sqrt{m}\right)$. By contrast, the VC bound provides an exponential decay on *m* [3,17]. Can the concentration bound in Theorem 4 be improved? The following proposition answers this question in the negative.

**Proposition** **4.**
*For any rational*
0<t<1
*, there exists a learning algorithm*
$L:\;Z^{m}\to H$
*, a distribution*
p(z)
*, and a parametric loss*
l:Z×H→[0,1]
*such that:*
$$p\left\{|R_{s}\left(h\right)-R\left(h\right)|=t\right\}=\frac{J(\hat{z};\;h)}{t},$$
*where the probability is evaluated over the random choice of*
s
*and the internal randomness of*
L
*.*


**Proof.** The proof is in Appendix F. □

Proposition 4 shows that, without making any additional assumptions beyond that of uniform generalization, the concentration bound in Theorem 4 is tight up to constant factors. Essentially, the only difference between the upper and the lower bounds is a vanishing $O(1/\sqrt{m})$ term that is *independent* of L.

## 5. Properties of the Learning Capacity

In this section, we derive bounds on the learning capacity under various settings. We also describe some of its important properties.

### 5.1. Data Processing

The relationship between learning capacity and data processing is presented in Lemma 1. Given the random variables x,y, and z and the Markov chain x→y→z, we always have J(x;z)≤J(x;y). Hence, we have a *partial order* on learning algorithms. This presents us with an important qualitative insight into the design of machine learning algorithms.

Suppose we have two different hypotheses $h_{1}$ and $h_{2}$. We will say that $h_{2}$ contains *less information* than $h_{1}$ if the Markov chain $s\to h_{1}\to h_{2}$ holds. For example, if the observations $z_{i}\in \left\{0,1\right\}$ are Bernoulli trials, then $h_{1}\in R$ can be the empirical average as given in Example 1 while $h_{2}\in \left\{0,1\right\}$ can be the label that occurs most often in the training set. Because $h_{2}=I\left\{h_{1}\geq m/2\right\}$, the hypothesis $h_{2}$ contains strictly less information about the original training set than $h_{1}$. Formally, we have $s\to h_{1}\to h_{2}$. In this case, $h_{2}$ enjoys a better *uniform* generalization bound because of data-processing. Intuitively, we know that such a result should hold because $h_{2}$ is less dependent to the original training set than $h_{1}$. Hence, one can improve the uniform generalization bound (or equivalently the learning capacity) of a learning algorithm by post-processing its hypothesis h in a manner that is conditionally independent of the original training set given h.

**Example** **6.**
*Post-processing hypotheses is a common technique in machine learning. This includes sparsifying the coefficient vector*
$w\in R^{d}$
*in linear methods, where*
$w_{j}$
*is set to zero if it has a small absolute magnitude. It also includes methods that have been proposed to reduce the number of support vectors in SVM by exploiting linear dependence [*
[30]
*], or some methods for decision tree pruning. By the data processing inequality, such techniques reduce the learning capacity and, as a consequence, mitigate the risk for overfitting.*


Needless to mention, better generalization does not immediately translate into a smaller true risk. This is because the empirical risk itself may increase when the hypothesis h is post-processed *independently* of the original training sample.

### 5.2. Effective Domain Size

Next, we look into how the size of the domain Z limits the learning capacity. First, we start with the following definition:

**Definition** **7**(Lazy Learning)**.**
*A learning algorithm*
L
*is called* lazy *if the training sample*
$s\in Z^{m}$
*can be reconstructed perfectly from the hypothesis*
h∈H*. In other words,*
H(s|h)=0*, where H is the Shannon entropy. Equivalently, the mapping from*
s
*to*
h
*is injective.*

One common example of a lazy learner is instance-based learning when h=s. Despite their simple nature, lazy learners are useful in practice. They are useful theoretical tools as well. In particular, because of the fact that H(s|h)=0 and the data processing inequality, the learning capacity of a lazy learner provides an upper bound to the learning capacity of *any* possible learning algorithm. Therefore, we can relate the learning capacity C(L) to the size of the domain Z by determining the learning capacity of lazy learners. Because the size of Z is usually infinite, we introduce the following definition of *effective* set size.

**Definition** **8.**
*In a countable space*
Z
*endowed with a probability mass function*
p(z)
*, the effective size of*
Z
*w.r.t.*
p(z)
*is defined by:*
${\mathrm{Ess}}_{p(z)}\;\left(Z\right)≐1+{\left(\sum_{z\in Z}\sqrt{p(z)\;(1-p(z))}\right)}^{2}$
*.*


At one extreme, if p(z) is *uniform* over a finite alphabet Z, then ${\mathrm{Ess}}_{p(z)}\;\left(Z\right)=\left|Z\right|$. At the other extreme, if p(z) is a Kronecker delta distribution, then ${\mathrm{Ess}}_{p(z)}\;\left(Z\right)=1$. As proved next, this notion of effective set size *determines* the rate of convergence of an empirical probability mass function to its true distribution when the distance is measured in the total variation sense. As a result, it allows us to relate the learning capacity to a property of the domain Z.

**Theorem** **5.**
*Let*
Z
*be a countable space endowed with a probability mass function*
p(z)
*. Let*
s
*be a set of m i.i.d. observations*
$z_{i}∼p\left(z\right)$
*. Define*
$p_{s}\left(z\right)$
*to be the empirical probability mass function that results from drawing observations uniformly at random from*
s
*. Then:*
$$E_{s}\;||p\left(z\right),\;p_{s}\left(z\right){\left|\right|}_{T}=\sqrt{\frac{{\mathrm{Ess}}_{p(z)}\;\left[Z\right]-1}{2\;\pi \;m}}+o\left(1/\sqrt{m}\right),$$
*where*
${\mathrm{Ess}}_{p(z)}\;\left[Z\right]$
*is the effective size of*
Z
*(see Definition 8).*


**Proof.** The proof is in Appendix G. □

A special case of Theorem 5 was proved by de Moivre in the 1730s, who showed that the empirical mean of i.i.d. Bernoulli trials with a probability of success ϕ converges to the true mean with rate $\sqrt{2\phi (1-\phi )/(\pi m)}$. This is believed to be the first appearance of the square-root law in statistical inference in the literature [31]. Because the effective domain size of the Bernoulli distribution, according to Definition 8, is given by 1+4ϕ(1−ϕ), Theorem 5 agrees with, in fact generalizes, de Moivre’s result.

**Corollary** **1.**
*Let*
$L:\;Z^{m}\to H$
*be a learning algorithm whose hypothesis is*
h∈H
*. Then,*
$J\left(\hat{z};\;h\right)\leq \sqrt{\frac{{\mathrm{Ess}}_{p(z)}\;\left[Z\right]-1}{2\;\pi \;m}}+o\left(1/\sqrt{m}\right)$
*. Moreover, the bound is achieved by lazy learners.*


**Proof.** Let $\tilde{h}$ be the hypothesis produced by a lazy learner. The simplest example is if h is equal to the training sample s itself. Then, we always have the Markov chain $s\to \tilde{h}\to h$ for any hypothesis h∈H. Therefore, by the data processing inequality, we have $J\left(\hat{z};\;h\right)\leq J\left(\hat{z};\;\tilde{h}\right)$. By Theorem 5, we have:
$$J\left(\hat{z};\;\tilde{h}\right)=\sqrt{\frac{{\mathrm{Ess}}_{p(z)}\;\left[Z\right]-1}{2\;\pi \;m}}+o\left(1/\sqrt{m}\right)$$
Hence, the statement of the corollary follows. □

**Corollary** **2.**
*For any learning algorithm*
$L:\;Z^{m}\to H$
*, we have*
$C\left(L\right)\leq \sqrt{\frac{|Z|-1}{2\;\pi \;m}}+o\left(1/\sqrt{m}\right)$
*.*


**Proof.** The function $f\left(p\right)=\sum_{z}\sqrt{p(z)(1-p(z))}$ is both concave over the probability simplex and permutation-invariant. Hence, by symmetry, the maximum effective domain size must be achieved at the uniform distribution p(z)=1/|Z|, in which case ${\mathrm{Ess}}_{p(z)}\;\left[Z\right]=\left|Z\right|$. □

### 5.3. Finite Hypothesis Space

Next, we look into the role of the *size* of the hypothesis space. This is formalized by the following theorem.

**Theorem** **6.**
*Let*
h∈H
*be the hypothesis produced by a learning algorithm*
$L:\;Z^{m}\to H$
*. Then:*
$$C\left(L\right)\leq \sqrt{\frac{H(h)}{2\;m}}\leq \sqrt{\frac{\log\left|H\right|}{2\;m}},$$
*where H is the Shannon entropy measured in nats.*


**Proof.** If we let I(x;y) be the mutual information between the r.v.’s x and y and let $s=\{z_{1},\;z_{2},\ldots ,z_{m}\}$ be the training set, we have:
$$\begin{array}{cc} I(s;\;h) & =H(s)-H(s\;|\;h)\\ \; & =\left[\sum_{i=1}^{m}\;H\left(z_{i}\right)\right]\;-\left[H\left(z_{1}|h\right)+H\left(z_{2}|z_{1},h\right)+\cdots \right] \end{array}$$Because conditioning reduces entropy, i.e., H(x|y)≤H(x) for any r.v.’s x and y, we have:
$$I\left(s;\;h\right)\geq \sum_{i=1}^{m}\;\left[H\left(z_{i}\right)-H\left(z_{i}\;|\;h\right)\right]=m\;\left[H\left(\hat{z}\right)-H\left(\hat{z}\;|\;h\right)\right]$$
Therefore:
(13)$$I\left(\hat{z};\;h\right)\leq \frac{I(s;\;h)}{m}$$
Next, we use *Pinsker’s inequality* [10], which states that for any probability measures *p* and *q*: ${\left||p\;,\;\;q|\right|}_{T}\leq \sqrt{\frac{D(p\;||\;q)}{2}}$, where ${\left||p\;,\;\;q|\right|}_{T}$ is total variation distance and D(p||q) is the Kullback-Leibler divergence measured in nats. If we recall that $J\left(s;\;h\right)={\left||p\left(s\right)\;p\left(h\right)\;,\;\;p\left(s,\;h\right)|\right|}_{T}$ while the mutual information is I(s;h)=D(p(s,h)||p(s)p(h)), we deduce from Pinsker’s inequality and Equation (Equation 13):
$$\begin{array}{cc} J(\hat{z};\;h) & =||p\left(\hat{z}\right)\;p\left(h\right)\;,\;\;p\left(\hat{z},h\right){\left|\right|}_{T}\\ \; & \leq \sqrt{\frac{I(\hat{z};h)}{2}}\leq \sqrt{\frac{I(s;h)}{2m}}\leq \sqrt{\frac{H(h)}{2m}}\leq \sqrt{\frac{\log|H|}{2m}}. \end{array}$$ □

Theorem 6 re-establishes the classical PAC result on the finite hypothesis space setting. However, unlike its typical proofs, the proof presented here is purely information-theoretic and does not make any references to the union bounds.

### 5.4. Differential Privacy

Randomization reduces the risk for overfitting. One common randomization technique in machine learning is differential privacy [32,33], which addresses the goal of obtaining useful information about the sample s as a whole without revealing a lot of information about any individual observation. Here, we show that differentially-private learning algorithms have small learning capacities.

**Definition** **9**([33])**.**
*A randomized learning algorithm*
$L:\;Z^{m}\to H$
*is*
(ϵ,δ)
*differentially private if for any*
O⊆H
*and any two samples*
s
*and*
$s^{\prime }$
*that differ in one observation only, we have:*
$$p\left(h\in O\;|\;s\right)\leq e^{\epsilon }\cdot p\left(h\in O\;|\;s^{\prime }\right)+\delta$$

**Proposition** **5.**
*If a learning algorithm*
$L:\;Z^{m}\to H$
*is*
(ϵ,δ)
*differentially private, then:*
$J\left(\hat{z};\;h\right)\leq \left(e^{\epsilon }-1+\delta \right)/2$
*.*


**Proof.** The proof is in Appendix H. □

Not surprisingly, the differential privacy parameters (ϵ,δ) control the uniform generalization risk, where small values of ϵ and δ lead to a reduced risk for overfitting.

### 5.5. Empirical Risk Minimization of 0–1 Loss Classes

Empirical risk minimization (ERM) of stochastic loss is a popular approach for learning from data. It is often regarded as the default strategy to use, due to its simplicity, generality, and statistical efficiency [1,3,13,34]. Given a fixed hypothesis space H, a domain Z, and a loss function l:H×Z→R, the ERM learning rule selects the hypothesis ${\hat{h}}_{s}$ that minimizes the empirical risk:(14)$${\hat{h}}_{s}=\arg\min_{h\in H}\;\left\{L_{s}\left(h\right)=\frac{1}{\left|s\right|}\sum_{z_{i}\in s}l\left(z_{i},h\right)\right\},$$
By contrast, the true risk minimizer $h^{⋆}$ is:(15)$$h^{⋆}=\arg\min_{h\in H}\;\left\{L\left(h\right)=E_{z∼p(z)}\;\left[l\left(z,\;h\right)\right]\right\}.$$
Hence, learning via ERM is justified if $L\left({\hat{h}}_{s}\right)\leq L\left(h^{⋆}\right)+\epsilon$, for some ϵ≪1. If such a condition holds and ϵ→0 as the sample size *m* increases, the ERM learning rule is called *consistent*.

Uniform generalization is a sufficient condition for the consistency of empirical risk minimization (ERM). To see this, we have by definition:$$\begin{array}{cc} E_{s}\left[L_{s}\left({\hat{h}}_{s}\right)\right] & =E_{s}\left[\min_{h\in H}L_{s}\left(h\right)\right]\\ \; & \leq \min_{h\in H}\left\{E_{s}\left[L_{s}\left(h\right)\right]\right\}=\min_{h\in H}L\left(h\right)=R\left(h^{⋆}\right), \end{array}$$
From this, we conclude that:$$\begin{array}{c} E_{s}R\left({\hat{h}}_{s}\right)-R\left(h^{⋆}\right)\leq E_{s}R\left({\hat{h}}_{s}\right)-E_{s}\left[L_{s}\left({\hat{h}}_{s}\right)\right]\leq C\left(L\right), \end{array}$$
where C(L) is the learning capacity of the empirical risk minimization rule. The last inequality follows from Theorem 2. In addition, because $R\left({\hat{h}}_{s}\right)-R\left(h^{⋆}\right)\geq 0$, we have by the Markov inequality:$$p_{s}\left\{R\left({\hat{h}}_{s}\right)-R\left(h^{⋆}\right)\geq t\right\}\leq \frac{E_{s}R\left({\hat{h}}_{s}\right)-R\left(h^{⋆}\right)}{t}\leq \frac{C(L)}{t}$$
Hence, the ERM learning rule is consistent if C(L)→0 as m→∞. Next, we describe when such a condition on C(L) holds for 0–1 loss classes. To do that, we begin with two familiar definitions from statistical learning theory.

**Definition** **10**(Shattered Set)**.**
*Given a domain*
Z*, a hypothesis space*
H*, and a 0–1 loss function*
l:Z×H→{0,1}*, a set*
$\left\{z_{1},\ldots ,z_{d}\right\}$
*is said to be shattered by*
H
*with respect to the function l if for any labeling*
$I\in {\left\{0,1\right\}}^{d}$*, there exists a hypothesis*
$h_{I}\in H$
*such that*
$(l\left(z_{1},h_{I}\right),\ldots ,l\left(z_{d},h_{I}\right))=I$*.*

**Example** **7.**
*Let*
Z=H=R
*and let the loss function be*
l(z,h)=I{z−h≥0}
*. Then, any singleton set*
{z}
*is shattered by*
H
*since we always have the two hypotheses*
$h_{0}=z-1$
*and*
$h_{1}=z+1$
*. However, no set of two points in*
Z
*can be shattered by*
H
*. By contrast, if the hypothesis is a pair*
(h,c)∈R×R
*and the loss function is*
l(z,h,b)=I{cz−h≥0}
*, then any set of two distinct examples*
$\left\{z_{1},z_{2}\right\}$
*is shattered by the hypothesis space.*


**Definition** **11**(VC Dimension)**.**
*The VC dimension of a hypothesis space*
H
*with respect to a domain*
Z
*and a 0–1 loss*
l:Z×H→{0,1}
*is the maximum cardinality of a set of points in*
Z
*that can be shattered by*
H
*with respect to l.*

The VC dimension is arguably the most fundamental concept in statistical learning theory because it provides a crisp characterization of learnability for 0–1 loss classes. Next, we show that the VC dimension has, in fact, an equivalence characterization with the learning capacity C(L). Specifically, under the Axiom of Choice, an ERM learning rule exists that has a vanishing learning capacity C(L) if and only if the 0–1 loss class has a finite VC dimension.

Before we establish this important result, we describe why ERM by itself is not sufficient for uniform generalization to hold even when the hypothesis space has a finite VC dimension.

**Proposition** **6.**
*For any sample size*
m≥1
*and a positive constant*
ϵ>0
*, there exists a hypothesis space*
H
*, a domain*
Z
*, and a 0–1 loss*
l:Z×H→{0,1}
*such that: (1)*
H
*has a VC dimension*
d=1
*, and (2) a learning algorithm*
$L:Z^{m}\to H$
*exists that outputs an empirical risk minimizer*
${\hat{h}}_{s}$
*with*
$J(\hat{z};\;{\hat{h}}_{s})\geq 1-\epsilon$
*.*


**Proof.** Let Z=X×Y, where X=[0,1] and Y={+1,−1} and let the loss be l(x,y,h)=I{y·(x−h)≤0}. In other words, the goal is to learn a threshold in the unit interval that separates the positive from the negative examples. Let x∈X be uniformly distributed in [0,1] and let $h^{⋆}$ be an error-free separator. Then, for any training sample $s\in Z^{m}$, the set of all empirical risk minimizers $\hat{H}$ is:
$$\hat{H}=\left\{h\in \left[0,1\right]:\;y_{i}=\mathrm{sign}\left(x_{i}-h\right),\;\forall i\in \left\{1,\ldots ,m\right\}\right\}$$
In particular, $\hat{H}$ is an interval, which has the power of the continuum, so it can be used to encode the entire training sample.Fix δ>0 in advance, which can be made arbitrarily small. Then, the probability over the random choice of the sample that $|\hat{H}|<\delta$ can be made arbitrarily small for a sufficiently small δ>0, where $|\hat{H}|$ is the length of the interval.Let $\hat{h}\in \hat{H}$ be a hypothesis that lies at the middle of $\hat{H}$, i.e.,:
$$\hat{h}=\frac{1}{2}\left[\arg\max_{x_{i}\in s\wedge y_{i}=-1}\;x_{i}+\arg\min_{x_{i}\in s\wedge y_{i}=+1}\;x_{i}\right]$$
Let $k=1+\log_{2}\left(1/\delta \right)$. Then, $\left[\hat{h}-2^{-k},\;\hat{h}+2^{-k}\right]\subseteq \hat{H}$ holds with a high probability (which can be made arbitrarily close to 1 for a sufficiently small δ). Let $\tilde{h}$ be a hypothesis whose binary expansion agrees with $\hat{h}$ in its first k+1 bits and encodes the entire training sample in the rest of the bits.Finally, the output of the learning algorithm is ${\hat{h}}_{s}$, which is given by the following rule:
If $\tilde{h}$ is an empirical risk minimizer, then set ${\hat{h}}_{s}=\tilde{h}$Otherwise, set ${\hat{h}}_{s}=\hat{h}$.Now, define the following *different* parametric loss $l^{\prime }:Z\to \left[0,1\right]$ to be a function that first uses ${\hat{h}}_{s}$ to *decode* the training sample s based on the coding method constructed above and, then, assigns 1 if and only if x∈s. To reiterate, this decoding succeeds with a probability that can be made arbitrarily high for a sufficiently small δ>0. Clearly, $l^{\prime }$ is a loss defined on the product space Z×H and has a bounded range. However, the generalization risk w.r.t. $l^{\prime }$ is, at least, equal to the probability that $|\hat{H}|<\delta$, which can be made arbitrarily close to 1. Hence, the statement of the proposition holds. □

Proposition 6 shows that one cannot obtain a non-trivial bound on the uniform generalization risk of an ERM learning rule in terms of the VC dimension *d* and the sample size *m* without making some additional assumptions. Next, we prove that an ERM learning rule *exists* that satisfies the uniform generalization property if the hypothesis space has a finite VC dimension. We begin by recalling a fundamental result in modern set theory. A non-empty set Q is said to be *well-ordered* if Q is endowed with a total order ⪯ such that every non-empty subset of Q contains a least element. The following fundamental result, which was published in 1904, is due to Ernst Zermelo [35].

**Theorem** **7**(Well-Ordering Theorem)**.**
*Under the Axiom of Choice, every non-empty subset can be well-ordered.*

**Theorem** **8.**
*Given a hypothesis space*
H
*, a domain*
Z
*, and a 0–1 loss*
l:H×Z→{0,1}
*, let ⪯ be a well-ordering on*
H
*and let*
$L:Z^{m}\to H$
*be the learning rule that outputs the “ least” empirical risk minimizer to the training sample*
$s\in Z^{m}$
*according to ⪯. Then,*
C(L)→0
*as*
m→∞
*if*
H
*has a finite VC dimension. In particular:*
$$C\left(L\right)\leq \frac{3}{\sqrt{m}}+\sqrt{\frac{1+d\log\frac{2em}{d}}{m}},$$
*where d is the VC dimension of*
H
*, provided that*
m≥d
*.*


**Proof.** The proof is in Appendix I. □

Next, we prove a converse statement. Before we do this, we present a learning problem that shows why a converse to Theorem 8 is not generally possible without making some additional assumptions. Hence, our converse will be later established for the binary classification setting only.

**Example** **8**(Subset Learning Problem)**.**
*Let*
Z={1,2,3,…,d}
*be a finite set of positive integers. Let*
$H=2^{Z}$
*and define the 0–1 loss of a hypothesis*
h∈H
*to be*
l(z,h)=I{z∉h}*. Then, the VC dimension is d. However, the learning rule that outputs*
h=Z
*is always an ERM learning rule that generalizes uniformly with rate*
ϵ=0
*regardless of the sample size and the distribution of observations.*

The previous example shows that a converse to Theorem 8 is not generally possible without making some additional assumptions. In particular, in the Subset Learning Problem, the VC dimension is not an accurate measure of the complexity of the hypothesis space H because many hypotheses dominate others (i.e., perform better across all distributions of observations). For example, the hypothesis $h^{\prime }=\left\{1,2,3\right\}$ dominates $h^{\prime \prime }=\left\{1\right\}$ because there is no distribution on observations in which $h^{\prime \prime }$ outperforms $h^{\prime }$. In fact, the hypothesis h=Z dominates all other hypotheses.

Consequently, in order to prove a lower bound for all ERM rules, we focus on the standard binary classification setting.

**Theorem** **9.**
*In any fixed domain*
Z=X×Y
*, let the hypothesis space*
H
*be a concept class on*
X
*and let*
l(x,y,h)=I{y≠h(x)}
*be the misclassification error. Then, any ERM learning rule*
L
*w.r.t. l has a learning capacity*
C(L)
*that is bounded from below by*
$C\left(L\right)\geq \frac{1}{2}\;{\left(1-\frac{1}{d}\right)}^{m}$
*, where m is the training sample size and d is the VC dimension of*
H
*.*


**Proof.** The proof is in Appendix J. □

Using both Theorems 8 and 9, we arrive at the following equivalence characterization of the VC dimension of a concept class with the learning capacity.

**Theorem** **10.**
*Given a fixed domain*
Z=X×Y
*, let the hypothesis space*
H
*be a concept class on*
X
*and let*
l(x,y,h)=I{y≠h(x)}
*be the misclassification error. Let m be the sample size. Then, the following statements are equivalent under the Axiom of Choice:*

H
*admits an ERM learning rule*
L
*whose learning capacity*
C(L)
*satisfies*
C(L)→0
*as*
m→∞
*.*

H
*has a finite VC dimension.*



**Proof.** The lower bound in Theorem 9 holds for all ERM learning rules. Hence, an ERM learning rule exists that generalize uniformly with a vanishing rate across all distributions only if H has a finite VC dimension. However, under the Axiom of Choice, H can always be well-ordered by Theorem 7 so, by Theorem 8, a finite VC dimension is also sufficient to guarantee the existence of a learning rule that generalize uniformly. □

Theorem 10 presents a characterization of the VC dimension in terms of information theory. According to the theorem, an ERM learning rule can be constructed that does not encode the training sample *if and only if* the hypothesis space has a finite VC dimension.

**Remark** **2.**
*One method of constructing a well-ordering on a hypothesis space*
H
*is to use the fact that computers are equipped with finite precisions. Hence, in practice, every hypothesis space is enumerable, from which the normal ordering of the integers forms a valid well-ordering on*
H
*.*


## 6. Concluding Remarks

In this paper, we introduced the notion of “ learning capacity” for algorithms that learn from data, which is analogous to the Shannon capacity of communication channels. Learning capacity is an information-theoretic quantity that measures the contribution of a single training example to the final hypothesis. It has three equivalent interpretations: (1) as a tight upper bound on the uniform generalization risk, (2) as a measure of information leakage, and (3) as a measure of algorithmic stability. Furthermore, by establishing a chain rule for learning capacity, concentration bounds were derived, which revealed that the learning capacity controlled both the expectation of the generalization risk and its variance. Moreover, the relationship between algorithmic stability and data processing revealed that algorithmic stability can be improved by post-processing the learned hypothesis.

Throughout this paper, we provided several bounds on the learning capacity under various settings. For instance, we established a relationship between algorithmic stability and the effective size of the domain of observations, which can be interpreted as a formal justification for dimensionality reduction methods. Moreover, we showed how learning capacity recovered classical bounds, such as in the finite hypothesis space setting, and derived new bounds for other settings as well, such as differential privacy. We also established that, under the Axiom of Choice, the existence of an empirical risk minimization (ERM) rule for 0–1 loss classes that had a vanishing learning capacity was equivalent to the assertion that the hypothesis space had a finite Vapnik–Chervonenkis (VC) dimension, thus establishing an equivalence relation between two of the most fundamental concepts in statistical learning theory and information theory.

More generally, the intent of this work is to bring to light a new information-theoretic approach for analyzing machine learning algorithms. Despite the fact that “ uniform generalization” might appear to be a strong condition at a first sight, one of the central claims of this paper is that uniform generalization is, in fact, a natural condition that arises commonly in practice. It is not a condition to require or enforce! We believe this holds because any learning algorithm is a *channel* from the space of training samples to the hypothesis space. Because learning is a mapping between two spaces, its risk for overfitting should be determined from the mapping itself (i.e., independently of the choice of the loss function). Such an approach yields the uniform generalization bounds that are derived in this paper.

It is worth highlighting that uniform generalization bounds can be established for many other settings that have not be discussed in this paper and it has found some promising applications. Using sample compression schemes, one can show that any learnable hypothesis space is also learnable by an algorithm that achieves uniform generalization [36]. Also, generalization bounds for stochastic convex optimization yield information criteria for model selection that can outperform the popular Akaike’s information criterion (AIC) and Schwarz’s Bayesian information criterion (BIC) [37]. More recently, uniform generalization has inspired the development of new approaches for structured regression as well [38].

## 7. Further Research Directions

Before we conclude, we suggest future directions of research and list some open problems.

### 7.1. Induced VC Dimension

The variational information $J(\hat{z};\;h)$ provides an upper bound on the generalization risk of the learning algorithm L across all parametric loss classes. This upper bound is *achievable* by the generalization risk of the *binary reconstruction loss*:(16)l(z,h)=I{p(z∈s|h)≥p(z∈s)},
which assigns the value one to observations z∈Z that are *more* likely to have been present in the training sample s upon knowing h, and assigns zero otherwise. In expectation, the generalization risk of this parametric loss is the worst generalization risk across all parametric loss classes.

Let both p(z) and p(h|z) be fixed; the first is the distribution of observations while the second is entirely determined by the learning algorithm L. Then, because the loss in Equation (Equation 16) is binary, it has a VC dimension, which we will call the *induced VC dimension* of the learning algorithm L [39]. Note that this induced VC dimension is defined for all learning problems, including regression and clustering, but it is *distribution-dependent*, which is quite unlike the traditional VC dimension of hypothesis spaces.

There are a lot of open questions related to the *induced VC dimension* of learning algorithms. For instance, while a finite VC dimension implies a small variational information, when does the converse also hold? Can we obtain a non-trivial bound on the induced VC dimension of a learning algorithm L upon knowing its uniform generalization risk $J(\hat{z};\;h)$? Along similar lines, suppose that L is an empirical risk minimization (ERM) algorithm of a 0–1 loss class that may or may not use an appropriate tie breaking rule (in light of what was discussed in Section 5.5). Is there a non-trivial relation between the VC dimension of the 0–1 loss that is being minimized and the induced VC dimension of the ERM learning algorithm?

### 7.2. Unsupervised Model Selection

Information criteria (such as AIC and BIC), are sometimes used in the unsupervised learning setting for model selection, such as when determining the value of *k* in the popular *k*-means algorithm [40]. Given that the notion of uniform generalization is developed in the *general* setting of learning, should the learning capacity C(L) serve as a model selection criterion in the unsupervised setting? Why or why not?

### 7.3. Effective Domain Size

The effective size of the domain of a random variable z in Definition 8 satisfies some intuitive properties and violates others. For instance, it reduces to the size of the domain |Z| when the distribution is uniform. Moreover, if z is Bernoulli, the effective domain size is determined by the *variance* of the Bernoulli distribution. Importantly, this notion is well-motivated because it determines the rate of convergence of an empirical probability mass function to its true distribution when the distance is measured in the total variation sense. As a result, it allowed us to relate the learning capacity to a property of the domain Z.

However, such a notion of effective domain size has some surprising properties. For instance, the effective size of the domain of two *independent* random variables is not equal to the product of the effective size of each individual domain! In rate distortion theory, a similar phenomenon is observed. Reference [10] explain this observation by stating that “ rectangular grid points (arising from independent descriptions) do not fill up the space efficiently.” Can the effective domain size in Definition 8 be motivated using rate distortion theory?

## Figures and Tables

**Figure 1 entropy-22-00438-f001:**
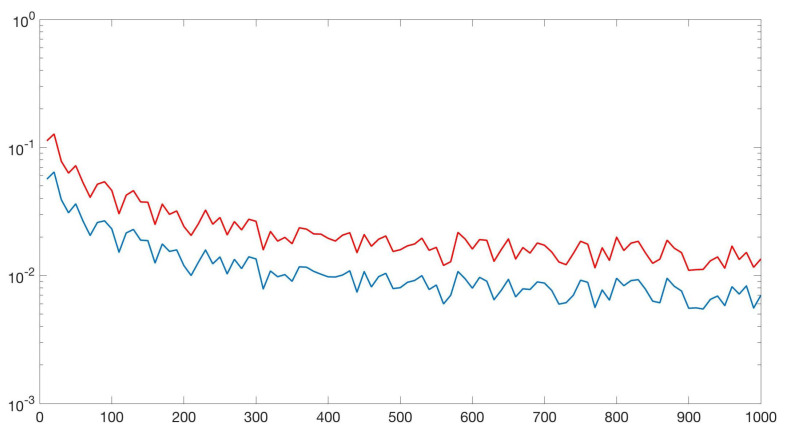
This figure corresponds to a classification problem in one dimension in which a classifier is a threshold between positive and negative examples. In this figure, the *x* axis is the number of training examples while the *y*-axis is the generalization risk. The red curve (top) corresponds to the difference between training and test accuracy when *z*-score normalization is applied before learning a classifier. The blue curve (bottom) corresponds to the difference between training and test accuracy when the data is not normalized.

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
