# Peer review of "Towards a Unified Theory of Learning and Information"

_entropy, 2020, doi:10.3390/e22040438_

Round 1
Reviewer 1 Report
In this paper, the author introduced the notion of “learning capacity" for algorithms that learn from data, which is analogous to the Shannon capacity of communication channels. It is demonstrated that this newly introduced notion can be used to develop generalization bounds for learning algorithms. Besides, the author also showed that it is closed related to algorithmic stability, information leakage, and data processing.
This is a well-written and well-polished paper. The problem studied in this paper is interesting and is important. The newly introduced notion "learning capacity" will draw some attention in machine learning as well as information theory.
Author Response
Response to Reviewer 1 Comments
I would like to thank the reviewer for taking the time to review the manuscript and for the valuable feedback.
Reviewer 2 Report
This paper is fantastic. I just have a few comments that might improve its reception amongst info theory-heavy, stat-learn-light folks like myself: -- I would not call the mutual information the TVD, and in fact, when you do use a proper mutual information later in Lines 342-343, you point out that it is a stronger condition than needed. The most problematic sentence is in the Intro with "where the mutual information is measured in the total variation distance"-- In Notation, you say you'll use calligraphic letters for alphabets, but then in Line 94, use a boldface letter
-- Line 151: "Our key contribution..." I'd say another key contribution is to point out that you should use TVD rather than mutual information.
-- I'm wondering how many assumptions you can weaken, what are the blocks to doing so. In particular: loss function maps to [0,1]; boundedness on the loss (DKL done weirdly sometimes diverges); data generated iid.
-- On the proof of Thm 2: it becomes a little unclear at the end. I think what you're saying is that if J<eps, then generalization risk < eps; and if generalization risk<eps for any loss function i.e. the uniform generalization risk, then you have J<eps. But you emphatically do not have that J=generalization risk -- Line 225: isn't another key here that TVD is bounded while MI can be infinite in those cases? -- Line 341, "compares well": can you be more specific? -- I'd stress the naturalness of uniform generalization not just in the Conclusion but in the Introduction. I think it's really great to have new definitions that allow you to really prove something, and those definitions should be highlighted!
Author Response
Response to Reviewer 2 Comments
I would like to take this opportunity to thank the reviewer for the valuable comments. Please see my response below.
- I will refrain from calling "variational information" the mutual information measured in the TV distance. Instead, I will refer to it in the introduction as the TV distance between the joint distribution and the product of marginals.
- Calligraphic letters are reserved for fixed sets (such as the domain X or the target Y). The sample s is random so it is denoted with a boldface letter. This will be clarified in the Notations section.
- In Line 151, I will rewrite it to say "Another key contribution" as suggested.
- The proofs assume that the loss has a bounded range. This seems to be difficult to remove as an assumption. The iid assumption is a standard PAC assumption and is crucial for many of the results (e.g. the finite hypothesis space setting, the equivalence with VC dimension, the countable domain setting, etc).
- If the domain has a finite dimension (so that the probability simplex is compact), then we have equality because the supremum is attained. The reason the proof uses epsilon is because the probability simplex may not be compact and the supremum may not be attained. In that case, we show that it can be attained arbitrarily well. The uniform generalization is itself defined as the supremum of generalization loss so equality follows after that.
- In Line 225, the comparison is still using the TV distance so it is bounded in both cases. The goal here is to say that measuring the variational information between h and z gives bounds that go to zero as the sample size goes to infinity even when the variational information between h and the entire sample s yields a vacuous bound (i.e. generalization gap =1).
- In Line 334, I will rewrite it to say that both bounds are equal to each other up to a multiplicative constant.
- I will stress on the naturalness of the definition in the Introduction as suggested.
Many thanks again for the valuable comments.
This manuscript is a resubmission of an earlier submission. The following is a list of the peer review reports and author responses from that submission.